# Medicinal Plant Preparations Administered by Botswana Traditional Health Practitioners for Treatment of Worm Infections Show Anthelmintic Activities

**DOI:** 10.3390/plants11212945

**Published:** 2022-11-01

**Authors:** Mthandazo Dube, Boingotlo Raphane, Bongani Sethebe, Nkaelang Seputhe, Tsholofelo Tiroyakgosi, Peter Imming, Cécile Häberli, Jennifer Keiser, Norbert Arnold, Kerstin Andrae-Marobela

**Affiliations:** 1Department of Bioorganic Chemistry, Leibniz Institute of Plant Biochemistry, Weinberg 3, D-06120 Halle (Saale), Germany; 2Department of Biological Sciences, Faculty of Science, University of Botswana, Gaborone P.O. Box 0022, Botswana; 3Kwame Legwame Traditional Association, Maun, Botswana; 4Institute of Pharmacy, Faculty of Natural Sciences, Martin-Luther-University Halle-Wittenberg, D-06120 Halle (Saale), Germany; 5Swiss Tropical and Public Health Institute, Kreuzstrasse 2, CH-4123 Allschwil, Switzerland; 6University of Basel, CH-4051 Basel, Switzerland

**Keywords:** neglected tropical diseases, traditional medicine, nematodes, schistosomiasis, soil transmitted helminths, *S. mansoni*

## Abstract

Schistosomiasis and soil-transmitted helminths are some of the priority neglected tropical diseases (NTDs) targeted for elimination by the World Health Organization (WHO). They are prevalent in Botswana and although Botswana has begun mass drug administration with the hope of eliminating soil-transmitted helminths as a public health problem, the prevalence of schistosomiasis does not meet the threshold required to warrant large-scale interventions. Although Botswana has a modern healthcare system, many people in Botswana rely on traditional medicine to treat worm infections and schistosomiasis. In this study, ten plant species used by traditional health practitioners against worm infections were collected and tested against *Ancylostoma ceylanicum* (zoonotic hookworm), *Heligmosomoides polygyrus* (roundworm of rodents), *Necator americanus* (New World hookworm), *Schistosoma mansoni* (blood fluke) [adult and newly transformed schistosomula (NTS)], *Strongyloides ratti* (threadworm) and *Trichuris muris* (nematode parasite of mice) in vitro. Extracts of two plants, *Laphangium luteoalbum* and *Commiphora pyaracanthoides*, displayed promising anthelmintic activity against NTS and adult *S. mansoni*, respectively. *L. luteoalbum* displayed 85.4% activity at 1 μg/mL against NTS, while *C. pyracanthoides* displayed 78.5% activity against adult *S. mansoni* at 10 μg/mL.

## 1. Introduction

Neglected tropical diseases (NTDs) include diverse groups of communicable diseases that globally infect or affect more than 2.7 billion of the most impoverished populations living in low- to middle-income countries of Africa, Asia and Latin America [1]. In Africa, 90% of the disease burden of NTDs is in Sub-Saharan Africa due to widespread poverty and the suitability of specific climates in Africa for some NTDs to thrive [2]. In Botswana, schistosomiasis and soil-transmitted helminth infections are some of the priority NTDs targeted for elimination [3,4]. Schistosomiasis and soil-transmitted helminth infections are prevalent in Botswana, particularly in the North-West Inland Wetland, the Okavango Delta, and along the Chobe River, both regions of major tourist attraction. Botswana has begun mass drug administration of albendazole in the hope of eliminating soil-transmitted helminths as a public health problem [4]. Although the prevalence of schistosomiasis does not meet the threshold required to warrant large-scale interventions, there has been an increase in the prevalence of *Schistosoma* species in Botswana between 2010 and 2019 [5], and it was predicted that there may be future epidemics of schistosomiasis based on the correlation between the flow of the rivers in the Okavango Delta and prevalence patterns of snail populations [6,7]. Mass drug administration may be an effective control measure against soil-transmitted helminth infections and schistosomiasis; however, the reliance on praziquantel for the treatment of schistosomiasis in mass drug administration control programs presents a constant threat of the development of drug resistance [8]. An example of this can be seen in Uganda, where the efficacy of praziquantel has been shown to be lower in schools that have a longer duration of mass drug administration [9]. The limited number of anthelmintic drugs and their prolonged use will inevitably lead to anthelmintic drug resistance. This is a well-documented issue in animal populations [10] and the resistance to anthelmintic drugs in veterinary species serves as a reference for how anthelmintic resistance may increase within the human population [11]. Mutations associated with benzimidazole resistance have been identified in eggs from human stools [12]. There is therefore a need for new anthelmintic drugs and the plants used in traditional medicine against parasitic helminths could provide promising leads.

The role of traditional medicine in health systems was re-affirmed when the WHO declared 2011–2020 as the Second Decade of African Traditional Medicine and extended this by formulating the WHO traditional medicine strategy 2014–2023. Traditional medicine plays an important role in the diagnosis, prevention or elimination of physical, mental and social illnesses. In Africa and the Diaspora, traditional medicine usage has been, and remains, a significant contributor to primary healthcare delivery [13,14].

In Botswana, the public healthcare system is made up primarily of modern biomedical formal structures. These include hospitals, clinics and outreach health posts and this ensures that every inhabitant lives no more than 15 km away from a health facility. In spite of their close proximity to modern biomedical structures, many people in Botswana still seek healthcare from traditional health practitioners. This is because it is familiar to them as it is ingrained in their culture and due to the fact that public health facilities are often overwhelmed and are not always well equipped. It can also be argued that access to traditional health practitioners is usually easier, faster and cheaper than finding a modern healthcare facility [15]. Seeking help from traditional health practitioners is also perceived as more personalized and confidential. An example of this is the fact that although there was an increase in the accessibility of combination anti-retroviral therapy, individuals with moderate and advanced HIV infection continued to use traditional medicine [16].

Traditional medicine is widely used in Africa in the management of worm infections and schistosomiasis [17,18,19,20]. Patient observations by traditional health practitioners therefore might lead us to botanicals useful for the treatment/management of worm infections and schistosomiasis. Previously, the main screening approaches used for the discovery of new anthelmintics were animal-based, target-based and phenotypic methods [21]. Animal-based methods are low throughput, time consuming and require a large amount of the extract or compound, and therefore in vitro screening methods (target-based and phenotypic) are now used [22]. Target-based methods screen extracts and compounds against one or more molecules with essential functions in the parasite metabolism [23]. These methods, however, do not consider the bioavailability of the compound to the parasite. Phenotypic screening employs whole parasites in vitro and quantitatively measures the phenotypic features after treatment of the parasites with extracts or compounds [21].

Natural products from botanicals might increase the available pool of potential new anthelmintics with unique structures and/or unique modes of actions, and hence delay or prevent resistance [24,25].

In this study, therefore, we documented traditional medical knowledge relating to worm infections and schistosomiasis in one area of high prevalence, the Ngamiland District in Botswana, and we determined the anthelmintic bioactivities of medicinal plants administered by local traditional health practitioners against a battery of hookworm, roundworm, nematode and *Schistosoma* species. To the best of our knowledge, this is the first comprehensive characterization of the anthelmintic bioactivities of traditional medicinal plants in Botswana.

## 2. Results

### 2.1. Traditional Medicinal Plants for Treatment/Management of Worm Infections

During our interaction with three traditional health practitioners, the medicinal plant uses of ten plant species were documented (Table 1).

With the exception of the *A. ferox* and *A. zebrina* species, preparations were made mostly from dried plant parts. The majority of the samples were leaves (55%), followed by roots (18%). Seeds, fruit and stem bark were each 9%. Roots are normally used in traditional medicine in Botswana since secondary metabolites are usually stored there, so it is interesting to note the majority of samples are leaves. Most of the samples are boiled (55%) and some are infused with cold, warm or hot water. Two samples are mixed with milk and this may be in order to reduce the amount of bioavailable compounds from the plant sample as milk is known to reduce the bioavailability of certain compounds. Milk, however, may also increase or decrease the excretion of certain metabolites in the extract [35]. The dosage for the majority of the samples was three times a day (60%) and only one sample (*S. panduriforme*) is taken once a day. To the best of our knowledge there is no supporting literature on the use of *B. albitrunca*, *C. pyracanthoides* and *L. luteoalbum* as anthelmintics, making this investigation the first to report their use to treat helminth infections. This is also the first in vitro bioactivity report on anthelmintic activity for *A. zebrina*, *T. sericea*, *C. mopane, B. albitrunca*, *C. pyracanthoides*, *S. panduriforme* and *L. luteoalbum*.

### 2.2. Bioactivities of Medicinal Plant Extracts against Parasites

Crude extracts prepared from the ten plant species were tested at the starting concentration of 100 μg/mL against the larvae (L3) of *Ancylostoma ceylanicum*, *Necator americanus*, *Strongyloides ratti*, *Heligmosomoides polygyrus* (adult and L3 larvae), *Trichuris muris* (adult worms) and NTS (Figure 1; Appendix A for numerical data). Only the most active extracts against NTS (threshold 70% at 50 μg/mL) (Figure 2) were further tested against adult *Schistosoma mansoni* beginning at 50 μg/mL (Figure 3). Activity above 50% at a concentration of 100 μg/mL for a plant extract is considered relevant [36,37]. Therefore, activity at a concentration of 100 μg/mL above 65% was considered ‘good’, between 30–65% ‘moderate’ and below 30% ‘weak’.

The leaves and roots of *L. luteoalbum* showed the highest activity against NTS with activity above 90% at 1 μg/mL. At 0.1 μg/mL, the activity was 28.6 ± 3.6% for *L. luteoalbum* leaves and 35.7 ± 0% for *L. luteoalbum* roots. *C. imberbe* leaves had 29.2 ± 4.2% activity, while *C. pyracanthoides* stem bark displayed 52.1 ± 2.1% activity. Both plant extracts were investigated at the test concentration of 10 μg/mL (Figure 2).

*C. pyracanthoides* stem bark (63.3 ± 6.9%) had the highest activity against *A. ceylanicum*, followed by *C. mopane* seeds (54.5 ± 1.9%) and *L. luteoalbum* leaves (53.1 ± 5.2%). The other plant extracts exhibited activity between 19.4 ± 4.2% and 45.6 ± 0.8%. *M. oleifera* leaves (58.2 ± 9.8%) exhibited the highest activity against *S. ratti* larvae. All plant extracts showed low activity against *N. americanus* larvae with activity between 0.3 ± 1.5% and 22.2 ± 2.7%. The plant extracts displayed activity between 31.7 ± 1.7% and 56.7 ± 3.3% against adult *H. polygyrus* worms but showed lower activity against the larvae (between 7.1 ± 3.3% and 23.2 ± 2%). *M. oleifera* leaves and *S. panduriforme* fruit had the highest activity (both at 56.7 ± 3.3%) against adult *H. polygyrus* worms, while *B. albitrunca* leaves (23.2 ± 2%) had the highest activity against *H. polygyrus* larvae. *S. panduriforme* fruit (66.7 ± 3.3%) was the only extract that showed activity above 65% against adult *T. muris* and it was therefore tested at a lower concentration of 50 μg/mL, at which it displayed reduced activity (29.6 ± 0%). The plant extracts generally showed the highest activity against NTS (between 69.2 ± 3.8% and 100 ± 0%, except for *A. ferox* leaves, 46.1 ± 3.8%, and *A. zebrina* leaves, 34.6 ± 3.8%) and they were tested at the reduced concentration of 50 μg/mL, where four plant extracts (*C. imberbe* seeds (98.1 ± 1.9%), *C. pyracanthoides* stem bark (76.9 ± 0%) and *L. luteoalbum* leaves (94.3 ± 1.9%) and roots (90.4 ± 1.9%)) displayed the highest activity. These four plant extracts were then tested at further reduced concentrations against NTS (Figure 2).

*C. pyracanthoides* exhibited the highest activity against adult *S. mansoni*, displaying 78.5 ± 0% activity at 10 μg/mL and 41.7 ± 1.7% activity at the lower concentration of 1 μg/mL. *C. imberbe* leaves displayed 78.4 ± 5.9% activity at 50 μg/mL, while *L. luteoalbum* leaves displayed 67.7 ± 3.9% activity also at 50 μg/mL (Figure 3).

## 3. Discussion

In this investigation, extracts from medicinal plants that are used as anthelmintics in traditional medicine by the traditional health practitioners from North-Western Botswana were tested for their in vitro anthelmintic activity against various helminth parasites. Conventional drug screening usually uses compound libraries to test for various biological activities in vitro, and the active compounds are then further tested in vivo. However, there is a very high failure rate with this approach, as often compounds that gave good in vitro activity can be inactive when tested in animal models [37] or can be toxic. About 60% of failures for potential new therapeutic drugs are due to lack of efficacy and toxicity, making these the two main causes for lack of success in drug development [38]. The approach of testing herbal extracts which are used in traditional medicine is referred to as reverse pharmacology [39]. The idea is that if a plant is already widely used in traditional medicine without reported toxicity, then the likelihood of the plant-based remedy being safe and effective is high [37]. Heinrich [40] stated, however, that working with a plant extract brings the major challenge that it is a mixture of active, partially active and inactive compounds whose activity is often not on a single target. Some of the compounds in the mixture can also be prodrugs, meaning that they need to be converted to an active form, usually by the gut microflora. Flavonoids are a group of compounds for which there is increasing evidence that they could possibly act as prodrugs. For example, flavonols are metabolized by the intestinal microflora to their corresponding hydroxyphenylacetic acids [41,42,43,44]. The presence of prodrugs may lead to lack of activity in in vitro assays as the conditions required for the prodrug to become active are not available. The biological activity of plant extracts can result from the overall effect of compounds with synergistic, additive or antagonistic activity, and this can lead to loss of activity when fractionation is done in an effort to isolate the active principle [45]. Studies have also shown that disease resistance is less likely to occur when an extract with many compounds is used rather than a single active compound [46,47]. Plenty of plants used in traditional medicine are now registered and marketed as botanical drugs in their crude form without having the active ingredients isolated [48]. A remarkable example is the phytopharmaceutical preparation Iberogast^®^, sold in Germany, which consists of nine different plant extracts. The preparation is used in the treatment of dyspepsia and showed therapeutic equivalence when it was compared with the synthetic drugs cisapride and metoclopramide, but it had fewer side effects than the synthetic drugs [49]. The plant extracts in our study are used by traditional health practitioners against helminths as extracts and were therefore tested in vitro as extracts. The plant extracts all showed anthelmintic activity to varying degrees.

The overall best anthelmintic activity was exhibited by *C. pyracanthoides* stem bark extract against *S. mansoni* adult worms, *A. ceylanicum* L3 and *N. americanus* L3. Only *S. panduriforme* fruit extracts had better activity against *T. muris* than *C. pyracanthoides* stem bark extract, while *B. albitrunca* leaves extract was the only plant extract with better anthelmintic activity against *H. polygyrus* L3 than *C. pyracanthoides* extract. Plant extracts and compounds with broad spectrum anthelmintic activity are desirable as multiple parasitic helminths are often endemic in the same regions. A plant extract showing anthelmintic activity against several target species across the Nematoda and Platyhelminthes phyla could be helpful in the isolation of compounds with the potential to be broad spectrum therapeutic agents [50]. *Commiphora* (Burseraceae) species are often used in traditional medicine in southern Africa for various ailments, including malaria where the stem is used and stomach aches where the bark, resin and leaf are used. Investigations have shown that *C. pyracanthoides* essential oil extract has various biological properties including anti-inflammatory (5-LOX enzyme inhibition), anti-cancer (against HT-29, MCF-7 and SF-268 cell lines), antimicrobial (*Bacillus cereus*) and antioxidant activity (ABTS and DPPH assays) [51]. *C. pyracanthoides* was not cytotoxic against kidney epithelial cells, indicating that it has selective activity against cancer cells.

*L. luteoalbum* leaves and root extract exhibited excellent activity against NTS with both the roots and leaves showing similar activity against the parasite. The plant originates from Europe and was introduced to southern Africa by early settlers, and it is now widespread and known as a winter weed in maize lands [52]. The plant has shown antifungal activity and the compounds responsible for the antifungal activity did not show cytotoxic activity when tested against the Vero cell line [53]. Although no cytotoxic effects were observed against the Vero cell line, further toxicological studies need to be done to ensure the safety of the plant. This is because plants sometimes produce toxic secondary metabolites to act as defense compounds against pathogens and herbivores [54]. The promising activity against schistosomes makes *L. luteoalbum* an ideal candidate for further investigations as it is a weed, meaning it grows easily even in unfavorable conditions and obtaining adequate biomass would not be a challenge [55].

*M. oleifera* leaves exhibited the best anthelmintic activity against adult *H. polygyrus* and *S. ratti* L3 larvae. *M. oleifera* is a fast-growing woody plant whose seeds, leaves and flowers have broad spectrum therapeutic applications [56]. Among the biological properties investigated, the leaves have shown anthelmintic activity reducing the worm burdens of *H. contortus*, *T. colubriforms* and *O. columbianum* in goats [33]. The *M. oleifera* leaf extract has also exhibited anthelmintic properties against *Trichuris* sp. and *Ostertagia* sp. The bioactive compounds thought to be responsible for the anthelmintic activity are heneicosane, di(2-ethylhexyl)phthalate (as 1,2-benzenedicarboxylic acid in [34]), heptacosane pentatriacontane and hexadecanoic acid ethyl ester [34].

*S. panduriforme* fruits had the highest activity against adult *T. muris* and also shared the best activity against adult *H. polygyrus* with *M. oleifera* leaves. *T. muris* has low cure rates using benzimidazole drugs and therefore new approaches are needed to eliminate morbidity from trichuriasis [50]. Further investigations could be carried out on the fruits of *S. panduriforme* in order to isolate the active principle. *S. panduriforme*, *C. imberbe* and *T. sericea* are some of the plants used by traditional health practitioners in the treatment/management of HIV-related opportunistic infections in Ngamiland District in Northern Botswana [57]. *S. panduriforme* and *T. sericea* are also used in South Africa for the treatment of tuberculosis and a study carried out by Green et al. [58] showed that *T. sericea* bark extract (MIC 25 μg/mL) had better activity against *Mycobacterium tuberculosis* than *S. panduriforme* leaves, although *S. panduriforme* leaves showed activity at a concentration higher than 100 μg/mL. The roots of *S. panduriforme* are also used for the treatment of oral diseases [59] and the plant is used for skin infections, wounds and ulcers [56]. The leaves of *S. panduriforme* have antiplasmodial activity against *Plasmodium falciparum*, although *Solanum nigrum* not *S. panduriforme* is reported to be used against malaria in traditional medicine [60].

The roots, leaves and bark of *T. sericea* are widely used in South Africa to prepare remedies used in ethnoveterinary medicine for the treatment of wounds, ticks and diarrhea [61]. *T. sericea* is a multipurpose medicinal plant used to treat many ailments and the plant contains various biological activities including anti-HIV, antifungal, antibacterial, antiparasitic, anticancer, wound-healing, antioxidant and anti-inflammatory activity [30].

*A. ferox* is used in traditional medicine in South Africa to treat intestinal worm infections [62] and several studies have shown the in vitro and in vivo anthelmintic activity of the plant against *H. contortus* and *H. gallinarum* [26,27,28]. *A. zebrina* is used in southern Africa for the treatment of myiasis and in vitro studies have shown that the leaf extract reduces pupation rate and pupal mass of *Lucilia cuprina* and *Chrysomya marginalis* [63]. *B. albitrunca* is widely used as a medicinal plant in southern Africa. Its uses include the treatment of constipation, diarrhea and epilepsy. In Botswana, it is used for the treatment of skin diseases, haemorrhoids and in ethnoveterinary medicine [64]. Antibacterial and antifungal activities have been reported for leaf and fruit extracts of *B. albitrunca* [65,66]. *C. mopane* is a dominant tree occurring in the dry regions of southern Africa, and it is used in traditional medicine for the treatment of tapeworms [31], syphilis, dysentery, diarrhea, inflamed eyes [67] and is also used in ethnoveterinary medicine [68]. *C. imberbe* is widely used in Africa to treat bacterial infections [69], sexually transmitted infections [70] and also in ethnoveterinary medicine [71]. The leaf extract of the plant has shown in vitro biological activity against *S. haematobium* [32], and anti-inflammatory activity [72]. Isolated compounds were active against *Mycobacterium fortuitum* and *Staphylococcus aureus* [73].

Overall, the plant extracts showed the best activity against NTS and only the best four were tested against adult *S. mansoni*. The plant extracts also showed moderate activity against adult *H. polygyrus* and adult *T. muris*. Activity of the plant extracts against *A. ceylanicum* L3 and *S. ratti* L3 was reduced. *N. americanus* and *H. polygyrus* L3 were the worms for which the plant extracts had the least effect with activity ranging between 0.3 and 23.2%. The anthelmintic activity shown by the plant samples shows the importance of traditional medicine, as traditional health practitioners have knowledge about which plants to use against various ailments. Further investigations need to be carried out in order to identify the active principles in the plant extracts and validate the safety of the plant extracts through toxicological studies. Once the active principles are identified, then crude extracts can be standardized with identified marker molecules.

## 4. Materials and Methods

### 4.1. Study Site

The study was conducted in the Ngamiland District in the North-West of Botswana, which is the site of a unique ecosystem, the Okavango Delta, where over 95% of its inhabitants depend on wetland resources to sustain their livelihoods. It is furthermore a hotspot for biodiversity in Botswana, which attracts traditional health practitioners to procure and use medicinal plants from the area. Historically, *S. mansoni* transmission is known to occur in the Okavango Delta due to the abundance of *Biomphalaria pfeifferi*, the snail intermediate host for the parasite [74]. Many Okavango Delta inhabitants from poorer sections of the society, and in rural villages around the Okavango Delta, are subsistence farmers who are engaging in flood recession farming (‘Molapo-Farming’). This farming practice utilizes wetland flooding patterns for planting. Unfortunately, the risk of exposure of farmers to *Schistosoma* is high as flood waters bring the snails serving as vectors. Usually, whole families are involved in Molapo-Farming, which has led in the past to schistosomiasis in school students, which seriously affected their performance. Two sites in the Okavango served as sampling origins, the Ngamiland District capital, Maun, and the village of Sehithwa, 80 km from Maun (Figure 4). Both villages are home to traditional health practitioners who collaborated with us in this study.

### 4.2. Study Design, Data Collection Methods and Ethical Considerations

The study design followed an exploratory, mixed-methods approach [75,76] to solicit traditional medical knowledge and therefore was of qualitative nature. Three traditional health practitioners collaborated with us in this study, the late Mrs. Tshwanelo Seputhe, Mr. Nkaelang Seputhe, both from Maun, and Mrs. Tsholofelo Tiroyakgosi based in Sehithwa. All three health practitioners have collaborated with one of us (K. Andrae-Marobela) for over fifteen years and have demonstrated reliable, in-depth traditional medical knowledge. Both female traditional health practitioners were/are involved in Molapo-Farming contributing to their livelihood, which is representative of many Okavango Delta inhabitants [77]. Initial traditional medical knowledge about worm infections was shared by the health practitioners during an ethno-survey undertaken between 2008 and 2010 [78], which included in-depth interviews and informal discussions conducted after obtaining community consent and individual prior informed consent. We also used informal conversations to discover categories of meaning [79] during the celebrations of the African Traditional Medicine Day in 2013 in Toteng, Ngamiland District, Botswana, which was organized by the Ministry of Health, Botswana. These data were supplemented by recent (January 2019), subsequent, semi-structured and unstructured conversations with the three traditional health practitioners to obtain more detailed knowledge and to confirm previously generated data to enhance credibility. Though conversations were unstructured, the interviewer had a guideline in mind to focus on knowledge of worm infections and schistosomiasis, but the idea was to let respondents express themselves freely on their own terms [80]. Data were also collected through informal participant observations of traditional health practitioners outside of their homes while accompanying them during medicinal plant collection. These participant observations provided nuances of subjective meaning and valuable narratives of health practitioners’ experiences. Ethical approval and a research permit were granted by the Ministry of Infrastructure, Science & Technology, Botswana (Permit no.: ETH 5 (1)), and by the Ministry of Health (Permit no.: PPME: 13/18/1 Vol VIII (354); HPDME 13/18/1).

### 4.3. Plant Collection and Extract Preparation

The traditional medicinal plants investigated in this study were collected together with the traditional health practitioners to avoid misidentification. The plant species were taxonomically identified using the dichotomous key in Coates Palgrave and Ellery and Ellery [81,82] and authenticated species were deposited in the University of Botswana Herbarium. Correct botanical names were counterchecked using the WFO plant list (www.wfoplantlist.org). The plant samples were air dried indoors at room temperature and ground to a fine powder using a blender, and 1 g of each powdered sample was extracted by sonication for 15 min with 10 mL of 80% MeOH (methanol and water have proved to be the solvents with the highest extraction efficiency, Ref [83]) at room temperature. The resulting solutions were evaporated to dryness under reduced pressure using a rotary evaporator maintained at 40 °C to afford crude extracts. A total of 10 mg of each crude extract was tested at the Swiss Tropical and Public Health Institute (Swiss TPH) to investigate their anthelmintic activities.

### 4.4. Anthelmintic Assays

In vitro studies using parasitic helminths were carried out in accordance with Swiss national and cantonal regulations on animal welfare under the permission number 2070. The anthelmintic assays to test the activity of the plant extracts against *A. ceylanicum* (zoonotic hookworm)*, H. polygyrus* (roundworm of rodents)*, N. americanus* (New World hookworm)*, S. mansoni* (blood fluke) [adult and newly transformed schistosomula (NTS)], *S. ratti* (threadworm) and *T. muris* (nematode parasite of mice) were carried out as described previously [84,85,86]. Three-week-old female NMRI mice were obtained from Charles River (Sulzfeld, Germany). Three-week-old female C57BL/6NRj mice and three-week-old male Syrian golden hamsters were purchased from Janvier Laboratories (Le Genest-Saint-Isle, France). Rodents were kept in types 3 and 4 macrolon cages under environmentally controlled conditions (temperature: 25 °C, humidity: 70%, light/dark cycle 12 h/12 h) and had free access to water (municipal tap water supply) and rodent food. Rodents were allowed to acclimatize for 1 week before infection. Statistical analysis (one way ANOVA, *p* ≤ 0.001) was performed using SigmaPlot 14.0.

#### 4.4.1. In Vitro Tests on *A. ceylanicum*, *H. polygyrus*, *N. americanus*, *S. ratti* and *T. muris*

The life cycles of the assayed nematodes are maintained at the Swiss TPH. Hamsters were infected per os with 140 *A. ceylanicum* L3 or subcutaneously with 150 *N. americanus* L3. The feces of infected hamsters were filtered to obtain *A. ceylanicum* and *N. americanus* eggs which were then cultivated on an agar plate for 8–10 days in the dark at 24 °C to obtain larvae (L3), while mice were used to obtain *H. polygyrus* larvae (L3) following the same procedure. *S. ratti* L3 were acquired as summarized by Garcia and Bruckner [87]. For the drug assay, 30–40 L3 were placed in each well of a 96-well plate for each extract. Larvae were incubated in 198 μL culture medium with the test samples at a concentration of 100 μg/mL. RPMI 1640 (Gibco, Waltham MA, USA) medium supplemented with 5% amphotericin B (250 μg/mL, Sigma-Aldrich, Buchs, Switzerland) and 1% penicillin 10,000 U/mL and streptomycin 10 mg/mL solution (Sigma-Aldrich) was used for the assays with *H. polygyrus* L3. Phosphate-buffered saline (PBS, Sigma-Aldrich) supplemented with 1% penicillin (10,000 U/mL) and streptomycin (10 mg/mL) solution was used to incubate *S. ratti* L3. *A. ceylanicum* and *N. americanus* L3 stages were incubated in Hanks’ balanced salt solution (HBSS; Gibco, Waltham MA, USA) supplemented with 10% amphotericin B and 1% penicillin (10,000 U/mL) and streptomycin (10 mg/mL) solution. Larvae were kept in the dark at room temperature for 72 h, except *A. ceylanicum*, which were incubated at 37 °C and 5% CO_2_, after which the effect of the extract was evaluated. For this, the total number of L3 per well was determined. Then, 50–80 μL of hot water (≈80 °C) was added to each well and the larvae that responded (the moving worms) were counted. The proportion of larval death was determined and the percentage of survival was determined by the ratio of moving larvae to the total number of larvae present in the well. The *N. americanus* L3 assay was an exception as the wells were stimulated by vigorous up and down pipetting. The in vitro tests on adult *H. polygyrus* were carried out by first infecting female NMRI mice with 88 *H. polygyrus* L3. Mice were dissected two weeks post infection and three hookworm adult pairs were placed in each well of a 24-well plate at a volume 1980 μL and exposed to the test extracts at a concentration of 100 μg/mL. For the in vitro assay with *T. muris* adult worms, female C57BL/6NRj mice were infected with 200 embryonated *T. muris* eggs. Seven weeks post infection, *T. muris* adult worms were collected from the intestines. Three *T. muris* adult worms were placed in each well of a 24-well plate containing 1980 μL culture medium and the test extracts at a concentration of 100 μg/mL. The adult worms of *H. polygyrus* and *T. muris* were scored microscopically based on their phenotype, using a viability scale ranging from 3 to 0 (3: good motility and no morphological changes; 2: low motility and light changes in morphology; 1: very low motility and morphologically impaired; and 0: death). In case the adult worms did not move enough for a clear scoring, they were stimulated with hot water at the last evaluation time point. The reference compounds were abamectin 10 μM (for *A. ceylanicum* L3), levamisole 10 μM (for *N. americanus* L3, *H. polygyrus* L3, *S. ratti* L3) and tribendimidine 10 μM (for adult *H. polygyrus* and adult *T. muris*). The negative control was 1% DMSO.

#### 4.4.2. In Vitro Tests Using *S. mansoni*

To obtain NTS, cercariae were collected from infected *Biomphalaria glabrata* snails (maintained at Swiss TPH) and were mechanically transformed. Briefly, 5–6 weeks post infection, infected snails were each placed in a single well of a 24-well plate. The snails were left under a neon lamp for 3–4 h. After removal of the snails, the plate was examined for cercariae. The cercariae were collected using a Pasteur pipette and the cercarial suspension was poured through a 100 μm filter into a 50 mL tube. The cercariae were transformed to NTS by placing 7 mL of cercarial suspension in a 10 mL syringe and connecting each syringe to a Luer Lok. After connecting another empty syringe to the opposite side of the Luer Lok, the liquid was pushed back and forth three to four times vigorously. The suspensions were then poured into 15 mL tubes and placed on ice in the dark for 7 min. The supernatants were removed and discarded by slowly pipetting, leaving the sedimented NTS. The NTS were kept in the incubator (37 °C and 5% CO_2_) in medium M199, supplemented with 5% FCS and 1% penicillin/streptomycin and 1% (*v*/*v*) antibacterial/antifungal solution39 until usage.

In order to obtain adult *S. mansoni*, cercariae were collected following the same steps mentioned above. The cercarial concentration was adjusted to 100 cercariae/100 μL and 100 μL aliquots were aspirated with a 1 mL syringe ensuring there were no air bubbles in the aspirates. The cercarial suspensions were then each injected subcutaneously into the necks of mice. After infection, the mice were kept at 25 °C with a 12 h day/night cycle. After 7 weeks post infection, mice were euthanized with CO_2_ for 5 min. The mesenteric veins of the infected mice were then dissected in order to collect adult *S. mansoni* worms. For adult *S. mansoni* and NTS, transparent flat-bottom 96- and 24-well plates were used, respectively (Sarstedt, Switzerland). A total of 30–40 NTS were incubated with the test extract (0.1, 1, 10, 50 and 100 μg/mL) in 198–199.8 μL of M199 medium (Gibco, New York, NY, USA) supplemented with 5% (*v*/*v*) FCS (Bioconcept AG, Allschwil, Switzerland), 1% (*v*/*v*) penicillin/streptomycin solution (Sigma–Aldrich, Buchs, Switzerland) and 1% (*v*/*v*) antibacterial/antifungal solution for up to 72 h at 37 °C and 5% CO_2_. The experiment was conducted in triplicate. For the adult *S. mansoni* assay, at least three worms (both sexes) were incubated in a final volume of 1980 μL–1998 μL RPMI 1640 supplemented with 5% (*v*/*v*) FCS and 1% (*v*/*v*) penicillin/streptomycin at 37 °C and 5% CO_2_ for 72 h and the test extract at 1, 10 and 50 μg/mL. The experiment was conducted in duplicate. Adult worms and NTS were judged via microscopic readout 72 h after incubation; they were scored according to motility, morphology and granularity (scores from 0 to 3) [84]. The reference compounds were auronofin 10 μM (for NTS) and praziquantel 10 μM (for adult *S. mansoni*). The negative control was 1% DMSO.

## 5. Conclusions

In this study, we report for the first time the use as anthelmintics and the in vitro anthelmintic activity of *B. albitrunca, C. pyracanthoides* and *L. luteoalbum.* We also report for the first time the in vitro anthelmintic activity of *A. zebrine, T. sericea, C. mopane* and *S. panduriforme.* The promising antischistosomal activity exhibited by *C. pyracanthoides* stem bark, as well as the leaves and roots of *L. luteoalbum*, warrant further investigation of the plants as potential sources of compounds with antischistosomal properties. The overall anthelmintic activity exhibited by the different plant species, especially *C. pyracanthoides*, require further investigations to identify the active anthelmintic principles and also to perform cytotoxicity studies to give evidence for the safety of the plants. Our investigation confirms the importance of indigenous knowledge and further interviews should be held with traditional health practitioners in order to tap into the vast knowledge of medicinal plants which they have. These plants could be promising leads for the discovery of much needed new therapeutic agents against soil-transmitted helminth infections and schistosomiasis. Additionally, the possibility of using the plants for veterinary applications may also be studied.

## Figures and Tables

**Figure 1 plants-11-02945-f001:**
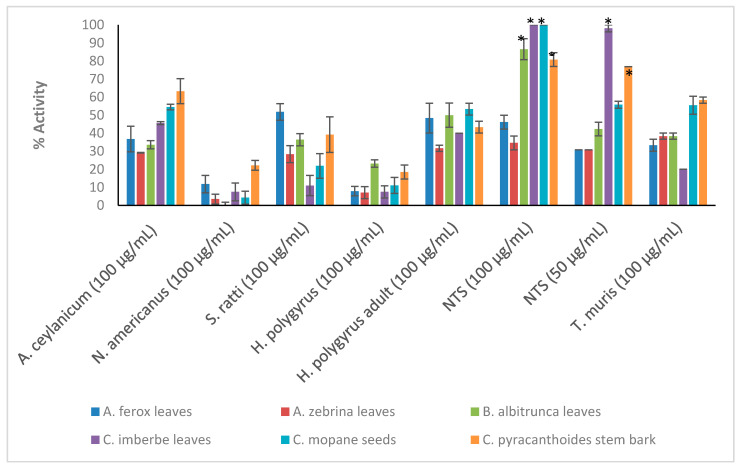
Anthelmintic activity of the plant extracts against the parasitic organisms. Activity based on three replicates. Statistical analysis (one way ANOVA, *p* ≤ 0.001, and an all pairwise multiple comparison procedure (Tukey Test), * extract activities determined to be significant) was performed using SigmaPlot 14.0 (Appendix A).

**Figure 2 plants-11-02945-f002:**
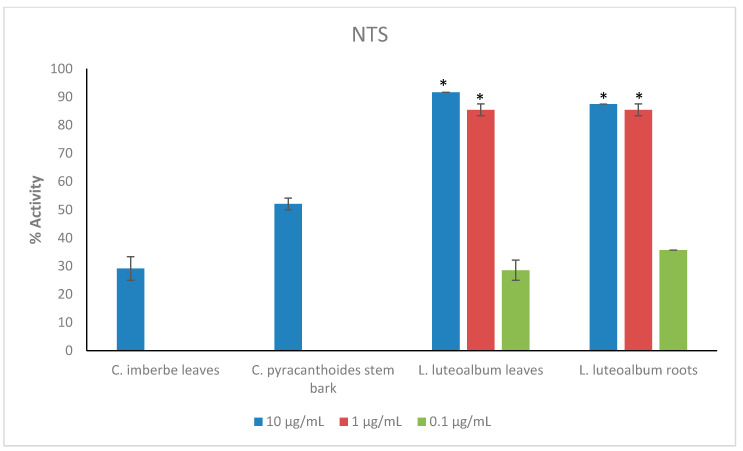
Anthelmintic activity of the most active plant extracts at reduced concentrations against newly transformed schistosomula (NTS). Activity based on three replicates. Statistical analysis (one way ANOVA, *p* ≤ 0.001, and an all pairwise multiple comparison procedure (Tukey Test), * extract activities determined to be significant) was performed using SigmaPlot 14.0 (Appendix A).

**Figure 3 plants-11-02945-f003:**
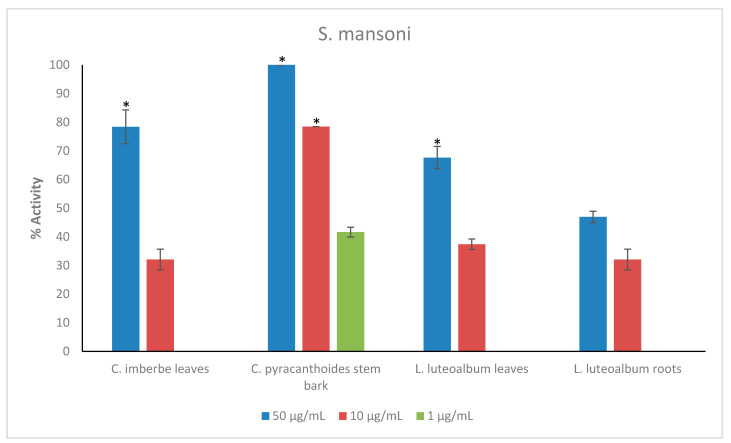
Anthelmintic activity of the plant extracts against *S. mansoni*. Activity based on three replicates. Statistical analysis (one way ANOVA, *p* ≤ 0.001, and an all pairwise multiple comparison procedure (Tukey Test), * extract activities determined to be significant) was performed using SigmaPlot 14.0 (Appendix A).

**Figure 4 plants-11-02945-f004:**
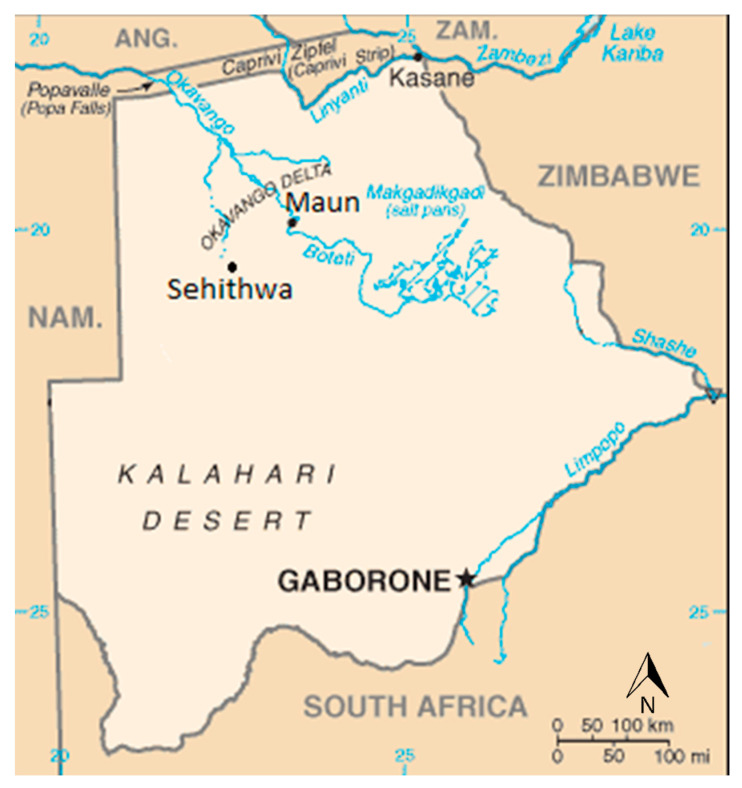
Map of Botswana showing areas where plants where collected (adapted from [74]).

**Table 1 plants-11-02945-t001:** Traditional medicinal anthelmintic plants used in North West District, Botswana.

Plant Species/Voucher Numbers	Plant Family	Local Name	Plant Part	Location	Preparation and Dosage	Supporting Literature
*Aloe ferox* Mill. (MDU/SEP-1/2019)	Xanthorrhoeaceae	-	Leaves	Maun	Pierce the leaves to extract the juice and infuse in cold or warm water. Take one cup twice a day after food.	In vitro anthelmintic activity against *H. contortus* and in vivo anthelmintic activity against *Heterakis gallinarum* [26,27,28]
*Aloe zebrina* Baker (MDU/SEP-2/2019)	Xanthorrhoeaceae	Ghopha	Leaves	Maun	Cut the leaves and infuse in water. Drink one cup two or three times a day.	The leaf extract is used against parasites and the root extract is used to treat bilharziosis (schistosomiasis) [29]
*Terminalia sericea* Burch. ex DC. (MDU/SEP-3/2019)	Combretaceae	Mogonono	Roots	Maun	Boil the roots and drink the water extract 3 times a day.	Roots are used to treat bilharzia (schistosomiasis) [30]
*Colophospermum mopane* (Benth.) Leonard (MDU/SEP-4/2019)	Leguminosae	Mophane	Seeds	Maun	Grind to a powder and put one tablespoon in a cup of hot water. Drink one cup three times a day.	Used in traditional medicine for the treatment of tapeworms [31]
*Boscia albitrunca* (Burch.) Gilg & Benedict (MDU/SEP-5/2019)	Capparaceae	Motopi	Leaves	Maun	Infuse one tablespoon of leaf powder in a warm cup of milk. Drink three times a day.	No reports
*Combretum imberbe* Wawra (MDU/SEP-6/2019)	Combretaceae	Motswere	Leaves	Maun	Boil the leaves and drink one cup of warm water extract three times a day.	Has shown in vitro biological activity against *Schistosoma haematobium* [32]
*Commiphora pyracanthoides* Engl. (MDU/SEP-7/2019)	Burseraceae	Moroka	Stem bark	Maun	Boil the bark and drink one cup two times a day.	No reports
*Solanum panduriforme* E. Mey. (MDU/SEP-8/2019)	Solanaceae	Tholwatholwane	Fruit	Maun	Mix the inner pulp of the fruit with milk. Three fruit pulps are mixed with one cup of milk. Drink the infused milk once a day.	Used in the treatment of tapeworm in conjunction with the roots of *Pseudeminia benguellensis* [29]
*Laphangium luteoalbum* (L.) Tzvelev (MDU/TIR-9/2019)	Compositae	Mookojane	Leaves	Sehithwa	Boil the roots and drink the water extract three times a day. After drinking the root extract, drink one cup of the boiled leaf extract at the end of the day.	No reports
*Laphangium luteoalbum* (L.) Tzvelev (MDU/TIR-10/2019)	Compositae	Mookojane	Roots	Sehithwa		No reports
*Moringa oleifera* Lam. (MDU/TIR-11/2019)	Moringaceae	Moringa	Leaves	Sehithwa	Boil the leaves and drink two cups of the water extract twice a day.	Anthelmintic activity against *H. contortus*, *Trichostrongylus colubriforms, Oesophagastum columbianum, Trichuris* sp. and *Ostertagia* sp. [33,34]

## Data Availability

Not applicable.

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
