# Peer review of "Medicinal Plant Preparations Administered by Botswana Traditional Health Practitioners for Treatment of Worm Infections Show Anthelmintic Activities"

_plants, 2022, doi:10.3390/plants11212945_

Round 1

Reviewer 1 Report (Previous Reviewer 3)

The authors have made some changes to the manuscript, but I am not convinced that this paper is suitable for publication in Plants.

1. The authors did not perform a proper statistical analysis. The current one is not adequate.

2. the graph caption that a statistical analysis was performed is not sufficient. We still do not know which data are significantly different.

3. The authors stated in the methods section that they used positive and negative controls. I would like to see the results of these controls in the graphs.

Author Response

Reviewer 2 Report (Previous Reviewer 4)

The Authors improved the article according to the comments.

Specific comments:

Figure 4: add North arrow

The sequence of affiliation should follow the same sequence of names

Author Response

Reviewer 3 Report (New Reviewer)

This is an interesting and important manuscript. I am impressed by the use of traditional healer interviews to identify candidate anthelmintics, an urgent need in the field of NTDs. Some of the results are very promising, others not so much so, which is reassuring in the context of screening for bioactive compounds or extracts. A few suggestions for improvement:

1. It would be worthwhile to include a brief introduction to current strategies for anthelmintic screening in the Introduction to put this work into context. Phenotypic screening such as done here is a primary strategy, and typically uses free-living stages of target parasites or surrogate species, both of which are used in this work. 

2. Throughout the manuscript, abbreviate the genus name after first appearance. 

3. Figure legends should include the number of Independent biological replicates and the number of replicates within an individual experiment. I suspect that only a single exposure to the extracts was done, which is a weakness, even if only a single extract preparation was made. It could have been tested in 3 distinct experiments for maximum value. In this same vein, some representation of statistical differences must be added to the bar graphs, denoting which values are significantly different from solvent controls and, preferably which are different among for a particular helminth within the experiment. The absence of statistical analyses is a weakness that needs to be corrected. This is particularly true when comparing inhibitory values; is (for example) 53% inhibition statistically different than 46%? If not, they must be treated as equal (non-different) values.  Values such as 0.3 % and 7% are almost certainly not statistically different than vehicle controls and this should be explicit in the text. This will require considerable modification of the Figures and the text.

4. Title of Figure 3: S. mansoni

5. Line 166: replace 'were' with 'at which'

6. Line 227: although lack of cytotoxicity against a single transformed cell line is certainly welcome, it is a very small aspect of toxicology, and it should be noted that many plant species make toxic compounds of many sorts. "Natural" by no means equals "safe", and it should be stressed that thorough toxicological testing should be done to establish safety. An additional reason to pursue chemical fingerprints for extracts of interest is to provide a basis for quality control should development continue.

7. Line 361 and following: see comment 3 above. More details of the assays should be added, because readers of Plants are, in general, unlikely to be familiar with anthelmintic screening. If a single Independent biological replicate was used, more justification for this experimental design should be added.

Round 2

Reviewer 1 Report (Previous Reviewer 3)

The authors explained my doubts. The manuscript may be published with minor corrections.

What DMSO concentration was used as a negative control? Pure DMSO would kill the parasites.

Authors should pay attention to minor editorial errors, e.g. formatting chemical formulas (CO2 and CO2).

Round 3

Reviewer 1 Report (Previous Reviewer 3)

The authors revised the manuscript as recommended. I believe it is suitable for publication on Plants.

This manuscript is a resubmission of an earlier submission. The following is a list of the peer review reports and author responses from that submission.

Round 1

Reviewer 1 Report

This study aims to investigate ten plant species with anthelmintic activity administered by Botswana traditional health practitioners.

The Methodology of the assays and the Results are very complete and well described, presented with five figures and one table, and the most important findings are well discussed and connected with other studies.

The results revealed that the plants tested could be promising leads for the discovery of new therapeutic agents.

The recommendation will be to accept the manuscript for publication in the present form.

Reviewer 2 Report

In the study of Boingotlo et althe traditional medical knowledge in relation to worm infections and schistosomiasis in Botswana was investigated, and determined the anthelmintic bioactivities of medicinal plants, administered by local traditional health practitioners, against a battery of hookworm, round-worm, nematode and Schistosoma species. The study, although interesting, it is preliminary and provides an initial characterization of the potential anthelmintic bioactivities of traditional medicinal plants in Botswana, which may be useful for future exploitation. Out of ten plant species used by traditional medicinal practitioners against worm infections that were tested in the study, two plants, Laphangium luteoalbum and Commiphora pyaracanthoides demonstrated promising anthelmintic activity against newly transformed schistosomules and adult S. mansoni respectively, which warrants further investigation.

Points for further consideration:

1.     Was the supplementary material provided? I could not locate it. 

2.     Provide information on the way anthelmintic activity was estimated I each case and what was used as a negative control. Were any positive controls included?

3.     In the description of the results, present the anthelmintic activity of the extracts as value % ± STDEV.

4.     No statistical analysis of the results was performed.

5.     Lines 138-139: “…while C. pyracanthoides stem bark displayed 52.1% activity both at at 10 μg/mL (Figure 2)”. This sentence is confusing and needs to be rephrased.

6.     Figure 4 is not necessary and can be omitted as it is not a direct outcome of the present paper. The names of the potentially responsible compounds in the M. oleifera leaf extract with the anthelmintic bioactivity and the reference of the paper are enough for this part of the discussion. 

7.     Further experiments elucidating the mode of action and the identification of bioactive compounds responsible for the observed bioactivity would enhance the significance and the value of the current findings. 

Reviewer 3 Report

The Authors of the manuscript tried to elaborate on an important topic, but not sufficiently in my opinion.

1) If water extracts are mainly used in the traditional treatment of parasite infections, why did the Authors use 80% methanol? The argument of high extraction efficiency is incorrect in this case.

2) Is traditional use for fresh or dried plants? Searching for drugs using "classic" phytochemical procedures is not wrong, but in this case there is no relation to the ethnopharmacological message.

3) Some information from paragraph 4.1. are more amenable to discussion.

4) I miss the chemical analysis of the extracts obtained (HPLC). How do the Authors know that the compounds shown in Fig. 4 were present in these extracts? What were their concentrations ???

5) What were the extracts dissolved in before being administered to the animals?

6) Were positive and negative controls used?

7) How was the statistical analysis performed?

I'm sorry, but for these reasons, I cannot recommend this manuscript for publication in Plants.

Reviewer 4 Report

The Authors investigated ten plant species used by traditional medicinal practitioners against worm infections and tested their effectiveness in in-vitro experiment. Although the topic is very interesting and well fitting the aim and the scope of the Journal, there are serious flaws that must be addressed in order to consider the present article for the publication.

However, suggestions for the improvement are provided below:

Title: “Medicinal plants with anthelmintic activity administered by 2 Botswana traditional health practitioners” I would encourage Author to give a more related title to the paper.

16 – as the first time appearing the paper, please provide full writing of the meaning of NTDS

17 – the keywords suggested are words already used in the title: medicinal plants; traditional health practitioners; anthelmintic activity; soil transmitted helminths. Find different keywords if possible.

Abstract – the abstract should also include main findings of the research, hopefully supported by some values that help the reader figure out if it is worthy to be read or not. Please add it.

102 – are 55 and 18% in weight or volume?

311-313 reporting the names of people in material methods is meaningless. Please describe the methodology avoiding direct personal references. People are not a methodology to report. The names of people who contributed to the research activity should be listed either as Authors or within acknowledgement section.

339- Mr. Bongani Sethebe is not a method nor a material. It is a person! Please report what kind of Dichotomous keys did he use for the authentication, instead.

342 – air dried at… what temperature? Did you use an oven or was it a natural drying?. In either case, the secondary metabolites of plants, which are rather sensitive to temperature, oxygen and light exposure would undergo spontaneous reactions which could dramatically affect the results of the experiment.

In material and methods – Please add the methodology applied to analyze the data. Did Authors perform any sort of statistical analysis?

Results – Results are provided without statistical analysis. Please add it

Supplementary material was not available